# Estimating Fluor Emission Spectra Using Digital Image Analysis Compared to Spectrophotometer Measurements

**DOI:** 10.3390/s23094291

**Published:** 2023-04-26

**Authors:** Hyeon-Woo Park, Ji-Won Choi, Kyung-Kwang Joo, Na-Ri Kim, Chang-Dong Shin

**Affiliations:** 1Center for Precision Neutrino Research, Department of Physics, Chonnam National University, Gwangju 61186, Republic of Korea; 2Department of Nuclear and Quantum Engineering, Korea Advanced Institute of Science and Technology (KAIST), Daejeon 34141, Republic of Korea; 3High Energy Accelerator Research Organization (KEK), Tsukuba 305-0801, Japan

**Keywords:** liquid scintillator, color space, digital camera, image analysis, spectrophotometer, fluor, emission spectrum

## Abstract

This paper describes a practical method for obtaining the spectra of lights emitted by a fluor in a liquid scintillator (LS) using a digital camera. The emission wavelength results obtained using a digital image were compared with those obtained using a fluorescence spectrophotometer. For general users, conventional spectrophotometers are expensive and difficult to access. Moreover, their experimental measurement setup and processes are highly complicated, and they require considerable care in handling. To overcome these limitations, a feasibility study was performed to obtain the emission spectrum through image analysis. Specifically, the emission spectrum of a fluor dissolved in a liquid scintillator was obtained using digital image analysis. An image processing method was employed to convert the light irradiated during camera exposure into wavelengths. Hue (H) and wavelength (W) are closely related. Thus, we obtained an H-W response curve in the 400~450 nm wavelength region, using a light-emitting diode. Another relevant advantage of the method described in this study is its non-invasiveness in sealed LS samples. Our results showed that this method has the potential to accurately investigate the emission wavelengths of fluor within acceptable uncertainties. We envision the use of this method to perform experiments in chemistry and physics laboratories in the future.

## 1. Introduction

A scintillator absorbs energy from external sources and emits it as light. A scintillation detector or scintillation counter can be constructed by combining a photomultiplier tube (PMT), a photodiode, or a silicon photomultiplier with a scintillator [1].

A liquid scintillator (LS) is a mixture of a solvent and fluor, which is a scintillating powder [2,3,4]. The most commonly used solvents are toluene, xylene, benzene, phenylcyclohexane, and triethylbenzene [5]. In addition, p-terphenyl (C_18_H_14_), 2-(biphenyl-4-yl)-5-phenyl-1,3,4-oxadi/azole (C_20_H_14_N_2_O, PBD), butyl-phenyl-bipheny-oxydiazole (C_24_H_22_N_2_O, butyl PBD), and 2,5-diphenyloxazole (C_15_H_11_NO, PPO) are the main fluors in use; as secondary wavelength shifters, 1,4-bis(5-phenyloxazol-2-yl)-benzene (C_24_H_16_N_2_O, POPOP) and 1,4-bis(2-methylstyryl)-benzene (C_24_H_22_, bis-MSB) are prevalently employed. Wavelength shifters are added to match the spectral sensitivity range of specific PMTs. In this study, the solvent, fluor, and secondary wavelength shifter in the LS were linear alkyl benzene (C_n_H_2n+1_-C_6_H_5_, *n* = 10–13, LAB), PPO, and bis-MSB, respectively. Furthermore, a PMT with a quantum efficiency (QE) of 400~450 nm was used. The incident wavelength was adjusted for the 400~450 nm ranges using an appropriate wavelength shifter [6,7,8].

Generally, a UV/Vis or fluorescence spectrophotometer is used to measure the absorption and emission spectra of LSs [9]. However, these devices are expensive and require careful handling to measure optical parameters. To overcome these limitations and simplify the related processes, a more accessible method was investigated in this study.

Information on fluorescent wavelengths can be obtained by taking a photo with a digital camera, or even using a camera on a mobile phone. When an image is taken with a camera, red, green, and blue (R, G, and B) color space values are stored. An image can also be represented based on the hue, saturation, and value (H, S, and V) of the color model. Once the RGB values are known, they can be converted into HSV values using the following equations [10,11,12,13]:
(1)$$\cos\theta \equiv \;\left\{\frac{\frac{1}{2}\left[\left(R-G\right)+\left(R-B\right)\right]}{\sqrt{{\left(R-G\right)}^{2}+\left(R-B\right)\left(G-B\right)]}}\right\}$$
(2)$$H=\left\{\begin{array}{c} \theta \\ 360-\theta \end{array}\begin{array}{c} \mathrm{if}\;B\leq G\\ \mathrm{if}\;B>G \end{array}\right.$$
(3)$$S=1-\frac{3}{\left(R+G+B\right)}\left[\min\left(R,G,B\right)\right]$$
(4)$$V=\frac{1}{3}\left(R+G+B\right)$$

Physically, hue (H) and wavelength (W) are closely related; thus, a hue–wavelength (H-W) relationship can be obtained [14,15].

Upon taking a picture of the light emitted from a certain sample, an H value is obtained, and it can be converted to a W value using the H-W relationship. Once the emission wavelengths of a sample have been obtained, their distribution can be determined. However, it is exceptionally difficult to precisely measure the wavelength down to a few nanometers. In the case of a spectrophotometer, an accurate fluorescence distribution can be obtained by increasing the wavelength by small increments, such as 1 nm. Image data are obtained by capturing photons during the camera’s exposure time. Therefore, the image data corresponding to each pixel represent an average of the entire wavelength region. In contrast, a fluorescence spectrophotometer measures the emission spectrum over a certain wavelength range. Then, the average value of the emission wavelength is calculated. In this study, this average value was compared with that of the image result.

Any uncertainties related to information loss during the RGB to HSV conversion process were also investigated. We analyzed whether or not the fluorescence spectrophotometer result was similar to that estimated from the image analysis within an acceptable error range. As already mentioned, most of the PMTs used in experimental high-energy physics have an optimized QE of around 400~450 nm. There is no need to distinguish the wavelength of light entering the PMT down to a few nanometers, although it is necessary to know its information in these wavelength regions. Therefore, we considered a method to easily identify the emission spectra of fluor by decoding digital photo images. In this study, the feasibility of using the image-based method to obtain average values without using a spectrophotometer was investigated.

In summary, there are several motivations for this paper. The first was to develop a new method to create a spectrometer based on a digital camera or even a mounted mobile phone camera. Secondly, by analyzing RGB digital images, the possibility of finding the fluor component dissolved in an LS sample could be investigated in the visible region. To date, there have been no such R&D studies elsewhere on the use of a liquid scintillator in particle detectors. Therefore, there is still room for improvement compared to using a conventional spectrophotometer. This idea shows its novelty and importance in cases in which a conventional spectrophotometer is inaccessible. Thirdly, this work will provide an inexpensive, non-invasive method of fluor component analysis in a sealed LS sample or detector. The proposed approach is not only interesting from the point of view of colorimetry techniques but can also have applied significance in the field of particle detector sensing research.

## 2. Measurement of Light Emissions from an LS Sample

### 2.1. Experimental Setup

A fluorescence spectrophotometer is commonly used to obtain LS emission spectra. The following steps are well known, not only among specialists in the field of spectroscopy, but also by users at technical universities for obtaining a fluorescence spectrum. First, the correct wavelength to be irradiated on the sample must be known. To this end, the absorption region must be measured using a UV/Vis spectrophotometer in advance. The wavelength at which no further absorption occurs is selected while scanning by varying the wavelength. Then, the sample is placed in the cuvette holder of the fluorescence spectrophotometer, as shown in Figure 1a. A fluorescence spectrophotometer irradiates monochromatic light, typically ultraviolet or visible light, at an energy that can be absorbed by the fluor. The fluor absorbs the excitation photons and excites the molecule from the ground state to the excited electron state. Subsequently, the molecule emits energy in the form of a photon when returning to its ground state. Next, to quantify the fluorescent emission at the fixed absorption wavelength, the emission spectrum is scanned in increments of a few nanometers while changing the wavelength. Thus, the fluorescence spectrum of the fluor in the LS is obtained, whereby the detector system analyzes the intensity of emitted light and determines the wavelength distribution [16,17].

The digital-camera-based imaging setup for the LS sample is shown in Figure 1b. A sample comprising bis-MSB fluor and PPO mixed in LAB solvent was synthesized. The LS sample was transferred to a transparent vial that was placed on a rotating board. Since the PPO absorption occurs below 380 nm, light with several 360 nm wavelengths was irradiated on the LS sample using a UV lamp. The board was rotated at various angles to eliminate the influence of the incident direction of the UV light on the measurement. The excitation light input to the sample was monitored using a high-resolution light sensor (THOR sensor, S120VC, Manufacturer: THORLABS, Newton, NJ, USA). Then, images were taken using a complementary metal oxide semiconductor (CMOS)-based digital single lens reflex (DSLR, Canon EOS D series, Manufacturer: Canon, Tokyo, Japan) camera. To avoid the influence of any external light, the images were captured in a dark room. A light sensor was installed in the dark room so that any stray light could be accounted for. The sensor output showed no sign of stray light. To further eliminate any background light, although the images were captured in a dark environment, a block board was used to surround the camera, considering the sensitivity of the camera lens. The images were obtained in the Joint Photographic Experts Group (JPEG) format. The JPEG image format entails a standard lossy compression method for still images based on down-sampling, discrete Fourier transformation, and encoding processes. Using the JPEG format, images can be stored in 256 color grayscale and represented in RGB form, making it a suitable file format for the image data analysis performed herein [18]. A digital camera measures the amount of light accumulated in a pixel during the exposure time. It is difficult to determine the wavelengths of all scanned regions based on an image obtained using a digital camera. Nevertheless, one of the advantages of image analysis is that various data can be used without any restrictions on location and time during the experimental processes.

**Figure 1 sensors-23-04291-f001:**
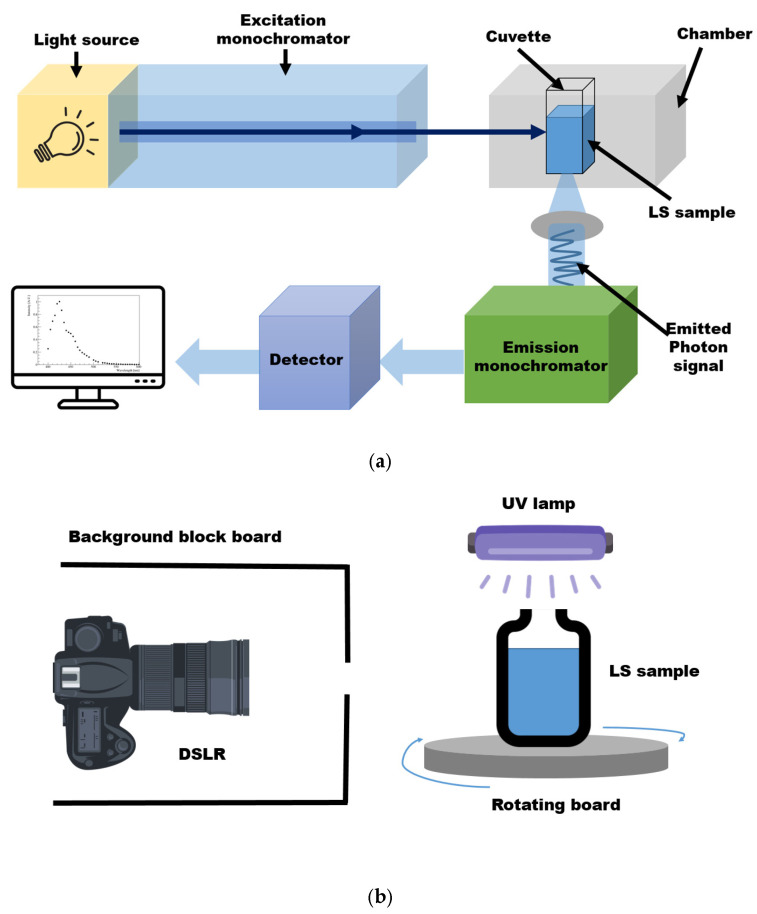
Schematics of the fluorescence spectrophotometer and digital imaging measurement setup. (**a**) In the conventional fluorescence spectrophotometer setup, monochromatic light is illuminated on the sample. The fluor absorbs energy and re-emits it as photons [19]. (**b**) The LS sample was illuminated by 360 nm light from a UV lamp, and digital images were taken. A black board with a hole was placed around the camera to avoid any background or stray light. The excitation light was filtered when captured by the camera. In order to minimize the effect at a specific angle, the sample on the board was rotated to obtain data at all angles.

For a clearer, more user-friendly explanation, the flow chart of each experimental device from the view point of spectrophotometer analysis and digital image analysis is shown in Figure 2. After synthesizing the LS, in the spectrophotometer analysis (2a–2e), a method using LED simulation was applied to obtain the average value of the fluor spectrum. In the image analysis (3a–3e), the average value of the data was obtained by capturing and shooting a moment in the fluorescent light emitted by the LS. Then, finally, the difference between the two results was compared.

### 2.2. H-W Relationship Using an LED

To obtain the H-W response curve for the 400~460 nm wavelength region, we generated light of the desired wavelength using an LED (Light Disc with 7 SMD RGB LED; Manufacturer: DFROBOT, Shanghai, China). The LED was controlled using an Arduino (Arduino Uno (R3); Manufacturer: Arduino, Ivrea, Italy) [20]. Table 1 shows that the desired wavelength was generated by inputting the corresponding RGB values. For example, if R (111), G (0), and B (154) were input into the Arduino, the color of a 400 nm wavelength was embodied. For calibration, we checked whether the desired wavelength was generated using both a color meter (TES-135A Color Meter; Manufacturer: TES Electrical Electronic Corp., Hsinchu County, Taiwan) and three laser modules with wavelengths of 405 and 440 nm.

Figure 3a shows a camera image of an LED-emitting light at a 438 nm wavelength; the area of the image was artificially divided into three regions (A1, A2, and A3) to remove the background to the maximum extent possible. The image was divided into three areas to determine the correlation between the V value and each area. The V value ranged from 0 to 1; when it was close to 0, the image was dark, whereas when it was close to 1, a lighter color was obtained [21]. When checking the V values of the image, the background was effectively removed at V values greater than 0.6. Only those pixels whose V value in the HSV model was greater than roughly 60% were selected to have the background removed. This boundary line was indicated as a rectangular box. The innermost box corresponded to a V value of approximately 0.6. Figure 3b plots the V values for each area. The V values were close to 0 in A3, while in A1, they were close to 1. Figure 3c shows the H distribution of the captured image of the LED. In the V distribution in Figure 3b, A3 had impure values; specifically, background black values mixed with those corresponding to actual colors. The values in A1 that were close to 1 were classified as signals. Thus, a representative H value for the wavelength of the LED was obtained. In this analysis, only those pixel regions with a V value exceeding 0.6 were selected. To obtain the representative H value, the mean H value was obtained according to the Gaussian fitting of the signal in Figure 3c. We assumed that the LED light was not monochromatic, but that, relatively, its emission bandwidth was not very large. Using this method, the H value of LED light as a function of wavelength was obtained from a camera image.

Figure 4 shows the H-W curve obtained using the LED light source for the 400~460 nm wavelength region. Specifically, we focused on the 400 nm wavelength region, because the fluors used in our study emit light in this wavelength range. In general, H tended to decrease as the wavelength increased, but this was not linear for the entire wavelength. In particular, wavelength separation in the 400~440 nm region became difficult due to the small change in H. There are some factors that affected the result; these include the investigation being performed with a different camera aperture, exposure time, saturation, and international organization for standardization (ISO) number [22]. Among them, the amount of light entering the camera was one of the most important factors. The thickness of the line in Figure 4 indicates uncertainties when the amount of light entering the camera was varied from 10 to 0.5 s. In addition, a great deal of care was taken to reduce any uncertainty caused by reflections on the sample.

### 2.3. Comparison of Results Obtained Using Spectrophotometer and a Digital Image

Figure 5 shows photographs of the light emitted from the LS samples. The samples were illuminated using an UV lamp at 360 nm. From left to right, samples comprising LAB, LAB + PPO, and LAB + PPO + bis-MSB are shown. LAB did not emit any light. Furthermore, it was almost impossible to visually identify a difference in the wavelengths of the samples with PPO and PPO + bis-MSB. As previously mentioned, PPO and bis-MSB are well-known fluors and are widely used for various neutrino experiments.

To compare the results obtained using the spectrophotometer and digital image, the average wavelength had to be determined from the data obtained using the spectrophotometer by averaging the entire emission spectrum. The black line in Figure 6 corresponds to the fluorescence distribution of the LAB + PPO + bis-MSB sample measured using a spectrophotometer (Varian Cary Eclipse Fluorescence Spectrophotometer; Manufacturer: Edinburgh Instruments, Edinburgh, UK). Specifically, the measurements were performed in the 400 to 600 nm ranges at intervals of 1 nm. The peak value of the distribution was obtained when the wavelength was 425 nm. This wavelength was chosen as a reference point, and the height of the peak at the maximum value was set as 1. From this curve, we wanted to extract the contribution of each R, G, and B component at each wavelength from 400 to 600 nm, and this was performed based on Table 1. Subsequently, the fluorescence value corresponding to this wavelength was normalized to rescale the R, G, and B values listed in Table 1. These newly obtained values were defined as the R-element, G-element, and B-element. For example, based on this method, at the wavelength of 425 nm, the original values of R (72), G (0), and B (218) could be rescaled to the R-element (0.32), G-element (0), and B-element (0.68), respectively. The sum of the (R, G, and B) elements must produce the fluorescence spectrum measured using the spectrophotometer, shown by the black solid curve in Figure 6. The rescaled (R, G, and B)-element components for other wavelengths can be obtained using the same method.

In Figure 6, the re-scaled (R, G, and B)-element values are plotted along with the fluor emission distribution. The average R, G, and B values of the fluorescence spectrum can be determined by integrating R, G, and B, respectively, over the entire wavelength region. All of the rescaled (R, G, and B)-element values were summed and represented as R-sum, G-sum, and B-sum values. Specifically, R-sum (47), G-sum (44), and B-sum (209) were obtained for this spectrum. This method is necessary because the spectrophotometer scans the entire area in increments of 1 nm, whereas the image captures a single time period. Therefore, it is necessary to predict the value in the same way. Once the (R, G, and B)-sum values were known, their values could be converted into HSV values, and finally, the wavelength could be estimated via the H-W relationship obtained using LED [14,15].

As shown in Table 1 and Figure 6, there are some differences between the monochromatic radiation of the excitation spectrum and radiation of a certain color (H, S, V) obtained by mixing the radiation of two different colors. In our case, radiation was imitated by using LED. This LED showed three separated (R, G, B) spectrums at each wavelength. We assumed that the LED-imitated radiation would have a similar transmittance efficiency in the RGB color filter to that of the real radiation, and its emission bandwidth was not very large.

Figure 7 shows the fluorescence distribution of a sample measured using a spectrophotometer and image analysis. The spectrophotometer results are shown as black dots in Figure 7a, and the red solid histogram represents the analysis result of the digital image shown in Figure 3. The blue-dotted histogram indicates the wavelength obtained from the R-sum, G-sum, and B-sum values determined from the fluorescence spectrum measurement. The average spectrophotometer result had a standard deviation of approximately 8 nm. Furthermore, the error estimation for this sample was less than 10 nm when photographed over the exposure time. Notably, the wavelength obtained from the image analysis and determined from the fluorescence spectrum agreed well within the uncertainty.

## 3. Summary

In this study, an alternative attempt was made to create a spectrometer based on a digital RGB camera. We measured the fluorescence emission spectrum of an analyte by means of digital image analysis, rather than by using spectrophotometer measurements. The fluorescence wavelengths obtained via image analysis were compared with the spectrophotometer results. Physically, H and W are closely related. Once the H value has been determined from image pixel analysis, the wavelength can be obtained. Thus, in this study, an H-W response curve was obtained by using an LED light source.

To apply the obtained response curve, we investigated the feasibility of determining the emission wavelength of a fluor dissolved in an LS. Through digital image analysis, the fluorescence wavelength was obtained after taking an image of the LS irradiated with a UV lamp. The expected wavelength of the fluorescence spectrum was determined using a spectrophotometer, and the wavelength obtained through image analysis agreed well within the estimated error. This method exhibits excellent potential to obtain the effective wavelength of a fluorescence spectrum through image analysis. Thus, we expect that our image analysis method can be applied to obtain the spectrum information of a completely unknown fluor at a reasonable level without even a rough idea of where its absorption or emission spectrum land is located.

## Figures and Tables

**Figure 2 sensors-23-04291-f002:**
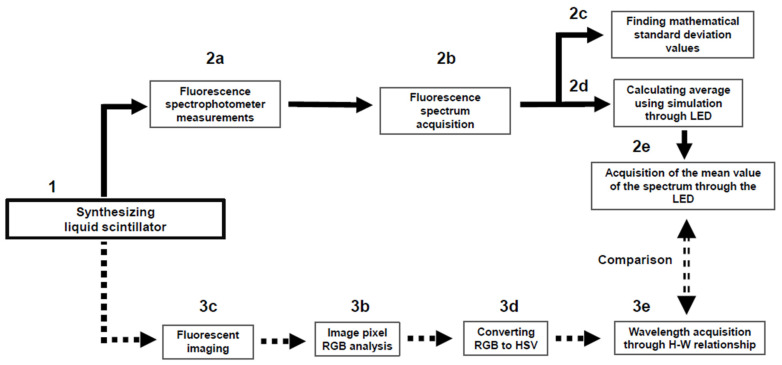
Flow diagram of spectrophotometer analysis (2a–2e) and digital image analysis (3a–3e). The final results between 2e and 3e were compared.

**Figure 3 sensors-23-04291-f003:**
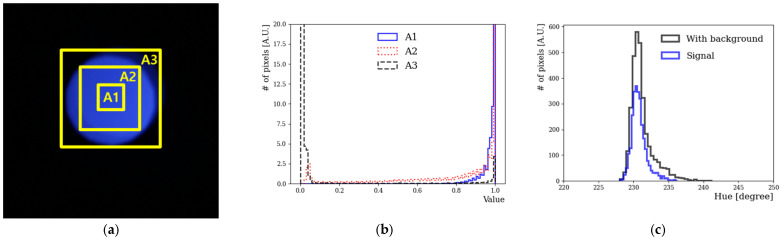
(**a**) Color images and the corresponding V and H values. The LED wavelength was 438 nm. The rectangular boxes represent the artificial division of the center of the image into three areas. (**b**) Distribution of V according to the area. (**c**) H distribution at 438 nm before and after background subtraction.

**Figure 4 sensors-23-04291-f004:**
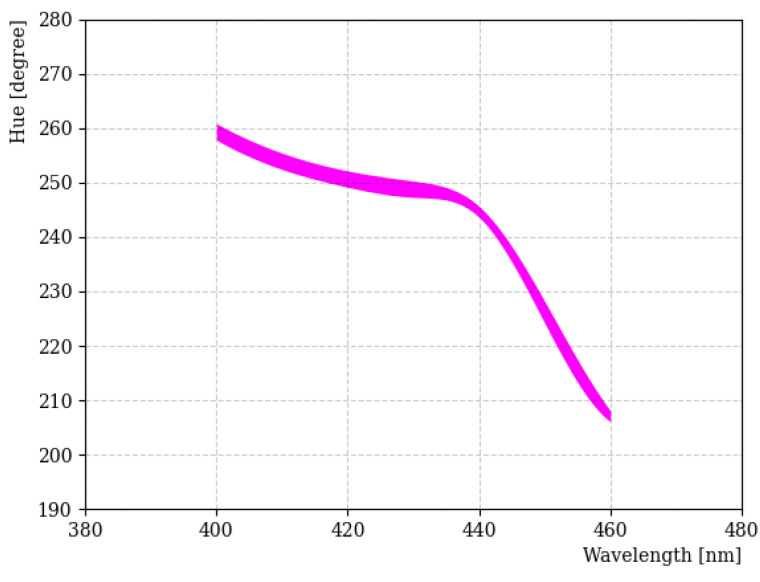
H-W curve obtained using the LED light source in the 400~460 nm region. The thickness of the line indicates the deviation of H across the limited dynamic range of the camera for these wavelengths.

**Figure 5 sensors-23-04291-f005:**
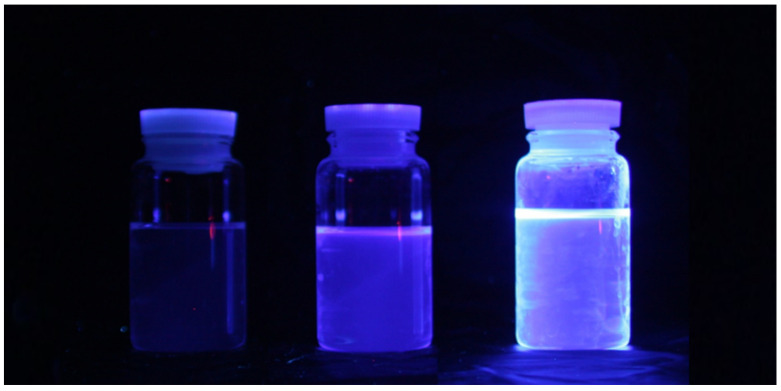
Light emitted from samples illuminated with a 360 nm UV lamp. From left to right, samples comprising LAB, LAB + PPO, and LAB + PPO + bis-MSB are shown. It is almost impossible to identify the wavelength difference with the naked eye.

**Figure 6 sensors-23-04291-f006:**
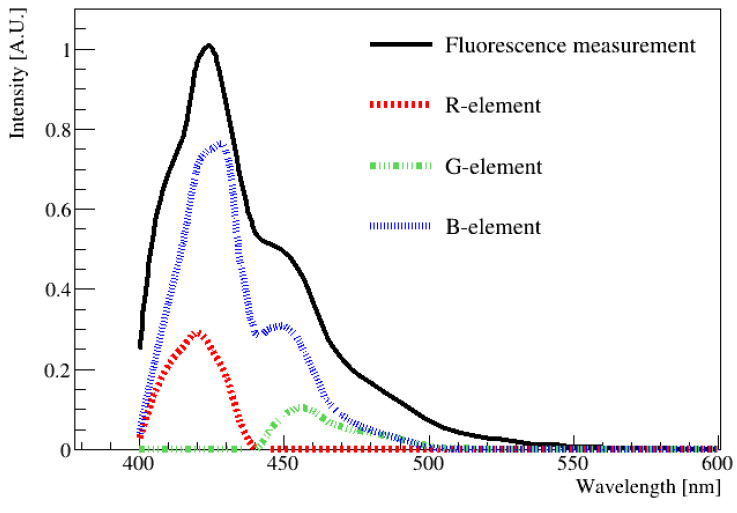
One example of an emission spectrum obtained using a spectrophotometer as a function of wavelength. The contribution of each component of R, G, and B in each wavelength of the solid black curve was extracted and re-scaled as R-element, G-element, and B-element, respectively. Then, re-scaled (R, G, and B) element values at each wavelength were drawn as a dotted red line, green line, and blue line. By combining (R, G, B) values, any wavelength in the visible region can be obtained. The peak value of the distribution is located at approximately 425 nm.

**Figure 7 sensors-23-04291-f007:**
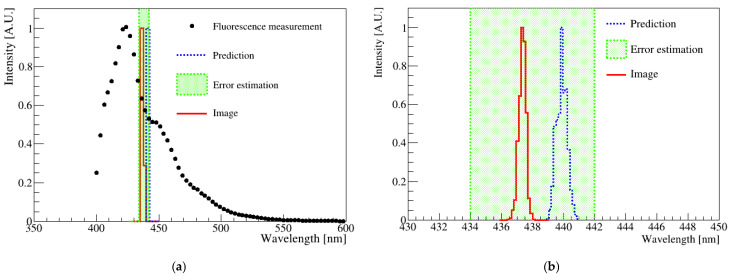
(**a**) Comparison plots of the spectrophotometer and digital image results. The points correspond to the spectrophotometer result, and the red solid histogram represents the digital image analysis result. The blue-dotted histogram represents the predicted average wavelength of the spectrophotometer via LED. The green-dotted band corresponds to the standard deviation of the mathematical mean of the spectrometer measurement results. (**b**) Magnification of the histogram area corresponding to the green box in (**a**). The two results were matched with the mathematical average within the standard deviation of the spectrophotometer.

**Table 1 sensors-23-04291-t001:** Wavelength of LED light based on R, G, and B values.

Wavelength (nm)	R	G	B
400	111	0	154
410	103	0	180
420	85	0	206
430	55	0	230
440	0	0	255
450	0	70	255
460	0	122	255
470	0	169	255
480	0	213	255
490	0	255	255
500	0	255	146
510	0	255	0
520	53	255	0
530	93	255	0
540	129	255	0
550	162	255	0
560	194	255	0
570	225	255	0
580	255	255	0
590	255	223	0
600	255	190	0

## Data Availability

Not applicable.

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
