# Peer review of "Estimating Fluor Emission Spectra Using Digital Image Analysis Compared to Spectrophotometer Measurements"

_sensors, 2023, doi:10.3390/s23094291_

Round 1

Reviewer 1 Report

The Reviewer's comments should be accompanied by detailed answers and indications of specific lines in the Manuscript where corrections have been made. 

The comments are mostly taken into account. The manuscript has been corrected and improved.

At the same time, the list of references has not been supplemented by modern sources of information.

Author Response

Please take a look at an attached file. Thanks.

Reviewer 2 Report

The paper “Estimation of fluor emission spectra using digital image analysis and its comparison with spectrophotometer measurements” proposed a method of obtaining the spectra of lights emitted by a fluor in a liquid scintillator (LS) based on a digital camera. The relevant advantage of the method described in this study is its non-invasiveness in a sealed LS sample.  There are some doubts:

1.“We analyzed whether the fluorescence spectrophotometer result was similar to that estimated from the image analysis within an acceptable error range.” (76-77) What is this acceptable margin of error mentioned in the article and how to determine it?

2. In section 2.2. How are the locations of zones A1, A2 and A3 determined? Are these areas fixed or artificially divided? Determine how these areas relate to background removal? In addition, select the area where the V value is greater than 0.6, what is the basis for this 0.6?

3. There is an experimental schematic diagram in this paper, but there is no physical diagram of the experimental device, can you provide and elaborate?

Author Response

(The authors gave the same response as above.)

Reviewer 3 Report

This paper describes a practical method of obtaining the spectra of lights emitted by a fluor in a liquid scintillator (LS) based on a digital camera. The document is well written, with minor English issues that should be corrected. However, there is still some work to do to clarify the reported results. Below you may find some questions I have that should be solved before the publication of the manuscript.

In Figure 2, what is the meaning of axis Y (# of events A.U.)? 

Line 153: Indicate technical details of the colorimeter.

In line 160, indicate that V is the "value" from the HSV format for clarity.

In line 170, it says that "LED light is not monochromatic, but its bandwidth emission is not very large" Could you provide an educated guess of its bandwidth, perhaps from the LED datasheet? 

Line 234: Typo, it should be "spectrophotometer."

Figure 6: Could you explain the difference between the blue dotted line and the red line? Should I understand that the spectrometer shows a peak wavelength at 425 nm while the image analysis shows that the main wavelength is around 437 nm? Does it mean there is an error of ~12 nm?

Following up on the questions above, Is the objective of this work only to have an averaged representation of the spectra of a sample? Or to get the full spectrum shape?

Does the rotating board in Fig. 1 have any effect on the measurements?

Additionally, I suggest these topics be discussed in the manuscript: Sources of uncertainty in the analysis: Integration time, lens distortion, imaging sensor responsivity (in the camera). 

Author Response

(The authors gave the same response as above.)

Round 2

Reviewer 3 Report

The authors have improved the manuscript and responded (in the text) to my questions.